# S-Adenosylmethionine (SAMe) for Central Nervous System Health: A Systematic Review

**DOI:** 10.3390/nu16183148

**Published:** 2024-09-18

**Authors:** Kyrie Eleyson R. Baden, Halley McClain, Eliya Craig, Nathan Gibson, Juanita A. Draime, Aleda M. H. Chen

**Affiliations:** School of Pharmacy, Cedarville University, Cedarville, OH 45314, USA; halleymcclain@cedarville.edu (H.M.); ecraig@cedarville.edu (E.C.); ngibson@cedarville.edu (N.G.); juanitaadraime@cedarville.edu (J.A.D.); amchen@cedarville.edu (A.M.H.C.)

**Keywords:** S-adenosylmethionine, central nervous system

## Abstract

**Background/Objectives:** S-adenosylmethionine (SAMe) is a natural compound used to improve mood-related symptoms. Our aim was to determine the efficacy, safety, and optimal dose of SAMe in Central Nervous System (CNS) signs (e.g., mood, behavior). **Methods:** We conducted a PRISMA-based systematic review by searching PubMed, CINAHL, and Web of Science using MeSH search terms. Articles were independently reviewed by two researchers (with a third resolving conflicts) during title/abstract screening and full-text review. Data were extracted in the same approach, with a quality assessment of included articles. **Results:** Out of 1881 non-duplicated studies, 36 were included in the review focusing on CNS signs (mood, behavior, sleep). Most studies (n = 32) achieved a 4 or 5 out of 5 points, indicating high study quality. Overall, SAMe was effective in 24 of 36 studies, with adverse events mostly consisting of mild, transient gastrointestinal disturbances. **Conclusions:** Many patients in these studies did experience improvements in CNS signs from using SAMe alone or in combination with existing therapy. However, future studies are needed to further understand the long-term effects of SAMe in the CNS.

## 1. Introduction

Mental health conditions are common, with a global burden of one in eight people affected according to the World Health Organization (WHO) [1,2]. Since the COVID-19 pandemic, the prevalence of depression and anxiety have risen approximately 28% and 26%, respectively [3]. Furthermore, the pandemic disproportionally affected females and younger individuals related to these central nervous system (CNS) conditions [3], yet the efficacy of standard treatment is suboptimal. The Mental Health America (MHA) 2024 report found that only 36% of youth with at least one major depressive episode said treatment helped “a lot” while 65% said it helped “some” [4]. Additionally, many agents used to treat CNS conditions may cause several adverse effects, such as weight gain, nausea/vomiting, and sexual dysfunction. Therefore, there is a need for better treatment options that are both safe and effective in improving patient outcomes.

S-adenosylmethionine (SAMe) is a nutraceutical marketed for its potential beneficial effects in several areas of the body, including the CNS. SAMe is produced in the liver from L-methionine and adenosine triphosphate (ATP) and is known for its role as a methyl donor in a variety of biological processes [5]. Some of these include DNA and RNA gene expression and neurotransmitter secretion, including dopamine, norepinephrine, and serotonin, which help elevate mood and support cognitive processes [5,6]. The replenishment of depleted neurotransmitters in CNS signs, like major depressive disorder, is important; however, the beneficial effects of SAMe may also be due to its anti-inflammatory properties. This may be explained, in part, by the ability of SAMe to synthesize glutathione, which aids in cellular detoxification through removal of free radicals [6].

The beneficial effects of SAMe on depression and other mood disorders are not fully established. However, low levels of SAMe have been reported in patients with major depressive disorder (MDD), while higher levels may lead to symptom improvement. Furthermore, as mentioned above, its role in the regulation of glutathione, polyamines, and monoaminergic neurotransmitters are just a few of the ways it may affect important processes in the brain [7]. Nevertheless, reports of SAMe use in CNS signs have varying results [8,9,10]. An updated review of the literature is needed to understand how SAMe affects specific conditions in the CNS and how these may inform dosing. Therefore, we conducted a systematic review to evaluate the safety, efficacy, and optimal dose of SAMe in CNS signs (e.g., mood, behavior, depression, anxiety, etc.).

## 2. Methods

PRISMA methodology was used for this systematic review, which the researchers followed in full compliance (see PRISMA Checklist in the Appendix A for full details) [11]. In collaboration with a research librarian, an initial search strategy was identified to address the research questions using the following MeSH search terms: “S-Adenosylmethionine” AND “Central Nervous System” OR “Behavioral Symptoms” OR “Anxiety” OR “Mood Disorders” OR “Sleep”. Based on these search terms, a research librarian refined the search in different search engines to maximize the accuracy and yield of the search in three databases: PubMed, CINAHL, and Web of Science. After confirmation from the research team, the research librarian finalized the search and ran the search with the dates of: 1 January 2004 to 17 April 2024. The search results were cleaned in Zotero (removing two retracted articles) and uploaded into Covidence (Melbourne, Australia). Covidence removed any duplicates identified.

Then, the research team began the review process. Three student research assistants and one faculty member were trained on the study protocol and Covidence system by the senior investigator (AC). The senior investigator also served as the project coordinator, checked for consistencies at each step, and resolved all disagreements and conflicts at each stage. All three phases were conducted in Covidence.

In the first phase, titles and abstracts were screened according to the (1) inclusion and exclusion criteria and (2) content. Inclusion criteria were as follows: research studies (randomized controlled trials, clinical trials, case studies or reports, cohort studies, prospective studies), English-language, and human studies. Exclusion criteria were as follows: review articles, guidelines, and expert opinion. Systematic reviews or meta-analyses also were excluded but were reviewed for any relevant articles. Secondly, articles were reviewed for content: all articles must include SAMe and a relevant CNS condition (mood, behavior, or sleep). Two research team members independently screened all titles and abstracts, categorizing them as “Yes”—meeting inclusion criteria, “No”—not meeting inclusion criteria, or “Maybe”—may meet inclusion criteria but needs further review. Any studies that were selected yes and/or maybe by two team members moved to full-text review. Any studies that were selected as no by two team members were excluded. Conflicts were resolved by a third researcher (AC).

In the second phase, full texts of all articles were pulled, read, and examined based on (1) criteria and (2) content. The same process with two independent reviewers and a third reviewer managing conflicts were followed. In this phase, the reason for exclusion was documented using a pre-specified list associated with the (1) criteria and (2) content. Any studies that were selected as “no” by two team members were excluded. All articles with “yes” were moved to the data-extraction phase.

In the third phase, a data-extraction template was built and utilized in Covidence to collect data to address the research questions. Reviewers also examined the quality of the study using the Mixed Methods Appraisal Tool (MMAT) (version 2018) [12], which allows for many different study designs to be evaluated for quality. Two research team members independently extracted the data and performed the quality appraisal, with the senior investigator reviewing all articles, resolving conflicts, and finalizing the data extraction into study tables. In addition, any missing data were noted in these tables. All research team members agreed on the final tables.

## 3. Results

A total of 2207 articles were pulled from the search (PubMed = 521, CINAHL = 173, Web of Science = 1513). After clean-up and removal of duplicates, 1882 articles underwent the review process. The PRISMA (Figure 1) overviews the study process, resulting in 36 articles that underwent data extraction.

### Study Characteristics

The total number of participants across the 36 articles [5,7,13,14,15,16,17,18,19,20,21,22,23,24,25,26,27,28,29,30,31,32,33,34,35,36,37,38,39,40,41,42,43,44,45,46] was 1799 (not including any systematic reviews). In the studies that specified gender, 60.1% of participants were female. The median study length was 8 weeks.

CNS-related signs included depression (general, mild, moderate, resistant, bipolar, non-remittent, subthreshold), anxiety, mood, suicide attempt, ADHD, cognitive deficits, 22q11.2 deletion syndrome depressive disorder, creatine transporter deficiency, and schizophrenia. Assessments used to determine efficacy included various mood-related tests, EEG mapping and psychometry, MRI, and plasma SAMe levels.

Table 1 provides a summary of the outcomes. When used in CNS-related conditions, SAMe dosing ranged from 200 to 3200 mg, with the most common dose being 800 mg. In many studies, SAMe was titrated up to the target dose and/or divided into two to four doses daily.

Table 2 provides a full description of remaining study characteristics, including the specific CNS sign studied, intervention(s), and measurements used. For the intervention, comparators were only included if applicable. These comparators included placebo, a different SAMe dose, an antidepressant with or without a placebo, or a different nutraceutical product (e.g., UP165).

Of the studies included, 33 assessed mood and depression (1 bipolar depression), 2 assessed anxiety, 1 assessed ADHD and cognitive deficits, and 1 assessed schizophrenia. Dosing strategies differed between CNS signs. More specifically, doses used for anxiety were 400 mg once or twice daily, 400 mg daily titrated to 800 mg twice daily for ADHD, and 400 mg daily titrated to 800 mg daily for schizophrenia. For depression, SAMe was dosed between 200 and 3200 mg daily, with studies titrating from lower to higher doses over a period of 3 days, 1 week, and 2 weeks.

The most common measurements of mood were versions of HAM-D (n = 14 studies), versions of CGI (n = 8 studies), and MADRS (n = 7 studies).

The compiled study outcomes for efficacy can be seen in Table 3 and safety in Table 4. If assessed by the study, all efficacy and safety data related to SAMe use were included. A total of 33 studies examined efficacy outcomes, while 29 studies included safety outcomes.

Improvements in scales or measurements of mood/CNS signs were seen in 24 studies. For studies that utilized HAM-D to assess SAMe, response rates were seen in up to 74% of participants and remission rates were seen in up to 93%. Six studies using the CGI tool (https://ia800200.us.archive.org/19/items/ecdeuassessmentm1933guyw/ecdeuassessmentm1933guyw_bw.pdf, accessed on 22 August 2024) found significant improvement with SAMe on these scores, while two studies showed no significant difference. For the MADRS tool, five studies reported that SAMe had no significant improvement, while the other two studies showed that SAMe had significant improvement.

Studies evaluating safety reported minimal adverse events, with gastrointestinal (GI) complaints identified as the most common (n = 10). In general, GI disturbances were reported as mild to moderate and transient. No studies reported severe GI side effects. Other adverse events reported were excitation, increased anxiety, mania/hypomania, sleep disruption, behavioral changes, headache, and fluid retention. One case report described an instance of medication-induced mood disorder in a patient taking an SSRI and SAMe combination. Another case report documented an attempted suicide. Nine studies reported no significant differences in adverse effects between SAMe and comparator, and one study reported fewer adverse effects in the SAMe group.

Study quality assessments can be found in Table 5. Most studies (n = 32) achieved 4 or 5 out of 5 points, indicating high study quality. A few studies (n = 4) had lower quality due to incomplete information as case reports, but they did not score any lower than 3 out of 5 points.

## 4. Discussion

Our findings suggest that SAMe shows potential therapeutic benefit in CNS-related signs, with major depressive disorder (MDD) being the most commonly studied among these. Improvements in mood were seen in a variety of validated scales, including HAM-D, MADRS, and CGI. Moreover, SAMe showed efficacy both as a monotherapy and as an adjunct to conventional antidepressants [7,24].

Nevertheless, results varied for SAMe use in the CNS. One study showed that it had no benefit on attention deficit/hyperactivity disorder (ADHD) symptoms [25]. Moreover, while some studies showed no significant difference between SAMe and a conventional antidepressant [5,7,24,34,36,41], others found it had no significant benefit compared to placebo [24,29,30,31,34,36,38,39]. As a comparison, a systematic review and network meta-analysis by Kishi et al. found that the efficacy of specific antidepressants (bupropion, escitalopram, vilazodone, etc.) was not better than placebo in preventing relapse at 6 months for adults with MDD [47]. Furthermore, while sertraline, vortioxetine, and desvenlafaxine had better efficacy, they were significantly more poorly tolerated compared to placebo. In children and adolescents, a meta-analysis by Rao et al. again noted that specific antidepressants (duloxetine, fluoxetine, venlafaxine, and escitalopram) were significantly more effective in treating depression compared to placebo [48]. However, of those agents, intolerability was also significantly higher in duloxetine and venlafaxine as well as several other antidepressants. Overall, choosing pharmacologic treatment for MDD is a delicate balance of weighing efficacy and safety. Because, in many cases, SAMe has demonstrated equivalence to antidepressants in terms of efficacy, it would be a valuable option for patients who wish to reduce or avoid adverse effects.

When considering SAMe as a treatment option, there are several factors to consider in an attempt to understand some of the varying results seen in the studies. Some of this variability may be explained by the differing dosing strategies, SAMe formulations, patient populations, and disease states. Furthermore, SAMe was studied in various combinations, such as with antidepressants (e.g., escitalopram) [14,20,33,37,44], *Lactobacillus* species [35,46], or other nutraceutical products [21,22,38,42]. Another factor to take into consideration is that SAMe may need time to reach its full effect for mood, similar to traditional antidepressants [21]. Finally, it is important to recognize that studies on MDD and other mood disorders have shown discrepancies in objective versus subjective assessments of mood and cognition [49,50]. Yet, many trial methodologies do not use both types of measurement in their design, making it difficult to make a comprehensive assessment.

To understand the most favorable dosage, it is important to note that SAMe is considered a hormetic nutrient because it demonstrates a biphasic dose response [51]. Hormesis is a biological phenomenon where exposure to low doses of a stressor induces adaptive, protective responses that enhance resilience against more severe stressors [52]. At low to moderate doses, SAMe exhibits potent antioxidant and anti-inflammatory properties because of its role in maintaining cellular redox homeostasis and modulating stress-response genes such as HO-1 [53]. The ability of SAMe to upregulate the Nrf2 pathway further highlights its crucial role in counteracting oxidative stress and neuroinflammation [54,55]. However, like other hormetic compounds, high doses of SAMe can be detrimental, leading to inhibition of antioxidant defenses and neurotoxicity. Thus, the therapeutic efficacy of SAMe hinges on careful dose regulation and balance.

While the optimal dose to optimize efficacy and minimize adverse effects of SAMe is not entirely known, benefits for mood and depressive symptoms were seen ranging from 200 mg to 3200 mg. Many studies started at lower doses and titrated to the target dose after a few days to 2 weeks or if they did not respond to the lower dose. Based on the available efficacy and safety from the included studies, we recommend considering doses starting at 200 mg to 800 mg for mild–moderate depression and 800 mg daily in one to two divided doses for resistant or non-remittent depression. Lower starting doses are warranted in patients with concomitant medications that may increase serotonin levels. Generally, we do not recommend a dose of 3200 mg due to higher rates of adverse effects. However, titrating to a target dose of 1600 mg per day is reasonable based on patient response. Finally, weight-based dosing with doses between 17 mg and 80 mg per kilogram per day may also be appropriate [23,26].

In addition to depression, SAMe showed benefit in other CNS conditions despite the limited amount of studies. When considering dosing for patients with anxiety, improvements in anxiety symptoms are seen with doses between 400 mg and 800 mg daily in one or two divided doses. Similarly, for patients with schizophrenia, improvements in aggression, depressive symptoms, and quality of life were seen at a dose of 400 mg daily titrated to 800 mg daily after 1 week. Interestingly, in the study by Green et al., SAMe improved depressive symptoms, although it did not improve ADHD symptoms [25]. This highlights that SAMe may have more consistent and substantial benefits when used in depression. In addition, the potential side effect of excitation caused by SAMe may limit its use in ADHD.

Furthermore, the oral or IM routes may be preferred over IV administration [5]. While the IV route has shown to be significantly worse in CNS diseases and should preferentially be avoided, it is still important to note that bioavailability may be highly variable between oral formulations [5].

Among the studies included, SAMe had a reasonable safety profile. Most common side effects of SAMe included mild to moderate gastrointestinal disturbances (constipation, abdominal discomfort, nausea) and headache. One study even reported fewer adverse events in the SAMe group [21]. Other adverse effects reported included increased anxiety [18,23,37], excitation [23], and fluid retention [36]. Rare adverse effects included anxiety [18,23,37], mania/hypomania [18,32], and psychomotor excitation [18]. Suicide attempt after SAMe use was documented with a single case report [19].

This systematic review has considerable strengths. First, PRISMA methodology was used, providing robustness to the study design as a systematic review. It also covers several different CNS-related signs, providing a more comprehensive and current assessment of the therapeutic potential of SAMe in these contexts. Furthermore, most studies (32 of 36) achieved a high study quality rating, allowing for more optimal evaluation of the evidence. Limitations of our review included the heterogeneity of study designs, populations, dosages, and outcomes measured. Because of this, it is difficult to generalize findings. In addition, our review process is not free from potential human error, as inconsistencies could be present in the application of inclusion/exclusion criteria. Finally, participant bias may be present, especially in subjective assessments of mood. However, many studies included an active or placebo comparator group to minimize this.

One of the current challenges of SAMe use in CNS health includes its lack of long-term efficacy and safety data past a 12-month period. Moreover, most evidence has been collected during 6 to 8 weeks, limiting our understanding of its chronic use. Another challenge is understanding the conditions for which it is most effective. While there is some evidence for its use in schizophrenia and bipolar disorder, in general, benefit was found mostly on depressive symptoms in these conditions. Thus, clinicians should be cautious in using SAMe for CNS signs outside of depression and anxiety.

Future studies are needed to assess long-term data of SAMe use on CNS health with and without concurrent antidepressant or nutraceutical therapy. In addition, issues in bioavailability between oral formulations must be addressed in order to provide accurate dosing recommendations and ensure optimal treatment. Thus, quality control in the manufacturing and production processes is necessary for safe and effective use of all supplements and nutraceuticals, including SAMe.

## 5. Conclusions

Current evidence suggests that use of SAMe improves depressive symptoms both as a monotherapy and with concurrent antidepressant or nutraceutical therapy. Clinicians should consider individual patient factors (e.g., specific disease treated, SAMe formulation and dose, patient comorbidities, medication history, goals of therapy, etc.) that could support the use of SAMe. While other therapies are currently used in first line management of CNS disease areas, SAMe may provide a useful option if they fail, are not appropriate, or if the patient prefers a nutraceutical approach. Future studies are needed to assess SAMe’s long-term efficacy and safety in depression, to further understand its effects alongside other therapies, and to address inconsistencies in oral formulations to ensure optimal dosing.

## Figures and Tables

**Figure 1 nutrients-16-03148-f001:**
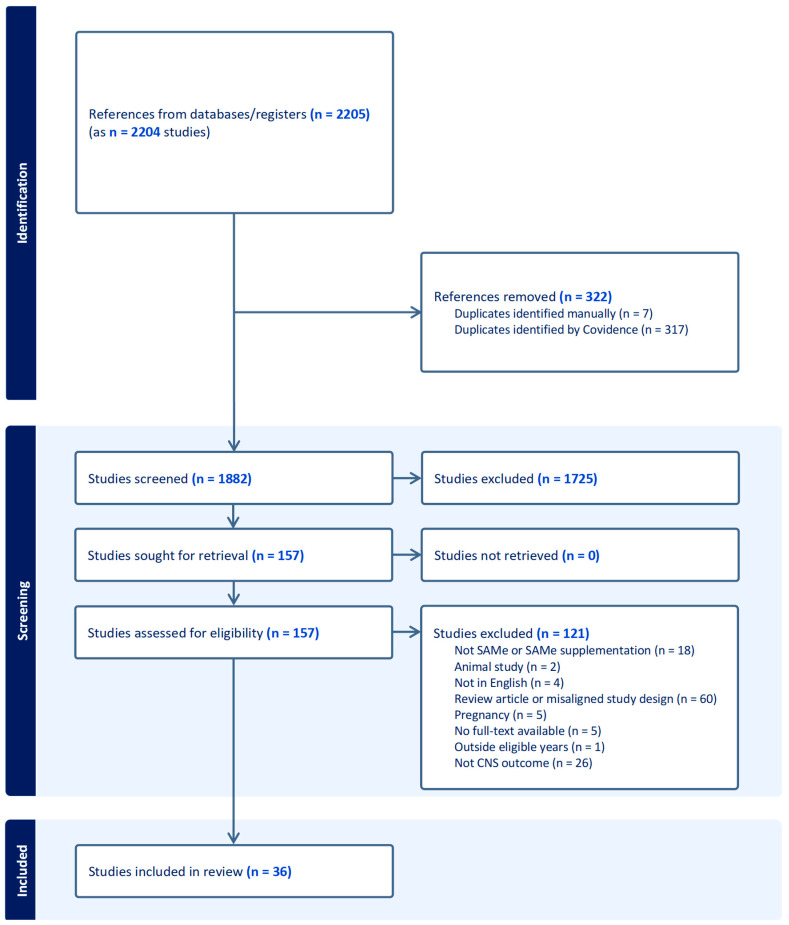
PRISMA [11] diagram overviewing study inclusion and exclusion process.

**Table 1 nutrients-16-03148-t001:** Overview of study findings related to safety and efficacy.

Condition	Efficacy Summary	Safety Summary	Dosing Ranges
CNS-Related Signs	24 of 36 studies reported positive findings (improvement in depressive symptoms) with SAMe There was a significant placebo effect overall—several studies found similar improvement with an SSRI, placebo, or SAMe ^2^	Case reports (n = 5): severe adverse events associated with SAMe use ^1^ Comparator-based studies (n = 18): typically did not find any significant difference in safety outcomes Most common adverse events were GI-related symptoms	Range: 200–3200 mg SAMe/day Many dose ranges titrated to 1600–3200 mg SAMe/day Usually in divided doses Often in combination with existing antidepressant regimen

^1^ The case reports did not use a Naranjo scale or other validated assessments to examine the likelihood of the cause and effect. ^2^ This indicates the variability of assessing change in depressive symptoms.

**Table 2 nutrients-16-03148-t002:** Characteristics of included studies examining the role of SAMe in CNS Signs.

First Author (Year) Study Design|Location N of Patients|Study Length	Intervention (with Dose) and Comparator	Disease (Sign)	Measurement of Mood/Depression
Abeysundera (2018) [13] Case report|Australia n = 1|2 weeks prior to incident	No dose given	Depression	Differential diagnosis and lab levels
Alpert (2004) [14] Open trial|United States n = 30|6 weeks	SSRI/Venlafaxine + SAMe: Initial: 400 mg twice daily At 2 weeks: 800 mg twice daily Comparator: None	Resistant Major Depressive Disorder	HAM-D–17, MADRS, CGI-I, CGI-S, SQ
Anderson (2016) [15] Case report|Canada n = 1|~1 month	SAMe: 400 mg twice daily Comparator: None	Anxiety (and hypothyroidism)	Not discussed
Arnold (2005) [16] Pharmacodynamic|Europe n = 12|15 days each medication + washout periods	SAMe: 1600 mg/day SAMe: 400 mg/day Comparator: placebo	Mood	EEG mapping and psychometry
Bambling (2015) [17] RCT|Australia n = 36|15 weeks	SAMe: 1600 mg/day SAMe: 800 mg/day	Major Depressive Disorder	BDI, ICD-DSM MINI, DASS, SCID, OQ45, WBS, QOLS
Carpenter (2011) [18] SR/MA|United States n = 14 studies on SAMe|N/A	SAMe: 500–1050 mg/day	Major Depressive Disorder	Varied
Chitiva (2012) [19] Case report|United States n = 1|4 days prior to event	No dose stated	Depression/suicide attempt	Not applicable
Cuomo (2020) [7] SR/MA|N/A n = 8 articles (1011 patients)|N/A	SAMe: 200–3200 mg/day	Major Depressive Disorder	Varied
De Berardis (2013) [20] Non-randomized experimental|Europe n = 25|8 weeks	Existing medication + SAMe: 800 mg/day	Major Depressive Disorder	HAM-D, CGI-I, SHAPS, SDS
Di Pierro (2015) [21] Open-label, randomized, observational|Europe n = 64 (60 completed)|12 months	Betaine 250 mg/day + SAMe: 500 mg/day Comparator: Amitriptyline 75 mg/day	Mild Depression	Zung Self-Rating Depression Scale
Djokic (2017) [22] RCT|Europe n = 60|3 months	Vit B complex + SAMe: 200 mg/day Comparator: placebo	Depression (mild to moderate)	HAM-D, CGI-S, CGI-I
Dolcetta (2013) [23] Non-randomized experimental|Europe n = 14|12 months	SAMe: 400–1600 mg/day; up to 80 mg/kg, depending on body weight and renal function	Mood Lesch–Nyhan Disease	N/A
Galizia (2016) [24] SR/MA|UK n = 8 studies (934 patients)|N/A	SAMe: 200–3200 mg/day	Depression	Varies
Green (2012) [25] RCT|Israel n = 12|6 weeks	SAMe: 400 mg/day titrated up to 1600 mg/day (800 mg twice daily) Comparator: placebo	22q11.2 deletion syndrome: depressive disorder, ADHD, cognitive deficits	Wechsler test: IQ, PANSS, YMRS, CGI-I, CDRS-R, ADHD-RS
Jaggumantri (2015) [26] Case report|Canada n = 2|Not described fully	SAMe: 50 mg/kg, with a safe and tolerable dose identified as 17 mg/kg/day	Creatine transporter (SLC6A8) deficiency	MRI Various assessments and questionnaires
Kalman (2015) [27] RCT|United States n = 34 in efficacy analysis (out of 42 enrolled)|8 weeks	SAMe: 400 mg/day Comparator: UP165 250 mg/day	Mild depression or anxiety	BDI-II, BAI, SOS-10
Levkovitz (2012) [28] RCT Re-Analysis|United States n = 55|6 weeks	SAMe Weeks 1–2: 800 mg/day Weeks 3–6: 1600 mg/day	Major Depressive Disorder	CPFQ
Limveeraprajak (2024) [5] SR/MA|N/A 23 trials (n = 2183)|N/A	SAMe: 200–1600 mg/day	Depressive symptoms	Varied
Mischoulon (2012) [29] RCT|United States n = 35|6 weeks	SAMe: 800–1600 mg/day Comparator: placebo	Major Depressive Disorder	HAM-D Plasma SAMe levels
Mischoulon (2014) [30] RCT|United States n = 189|12 weeks	SAMe: 1600–3200 mg/day Escitalopram: 10–20 mg/day Comparator: placebo	Major Depressive Disorder	HAM-D
Murphy (2014) [31] RCT|United States n = 20 (17 completed)|6 weeks	SAMe: Week 1: 800 mg/day Week 2: 400 mg/day Week 3: 800 mg/day Week 4: 1600 mg (only 3/7 days of week)	Persistent Treatment-Refractory Bipolar Depression	HAM-D, MADRS, YMRS
Olsufka (2017) [32] Case Report|United States n = 1|~1 week	SAMe: 400 mg/day for 3 days then increased to 800 mg/day (up to day 10)	Depression	Not applicable
Papakostas (2010) [33] RCT|United States n = 73 (55 completed)|6 weeks	Antidepressant + SAMe: 800 mg/day (up to 1600 mg/day) Comparator: antidepressant + placebo	Major Depressive Disorder	HAM-D, CGI-S
Peng (2024) [34] SR/MA|Taiwan n = 14 studies (1522 patients)|N/A	SAMe: 200–3200 mg/day	Depression	Varies
Saccarello (2020) [35] RCT|Europe n = 89|6 weeks	*Lactobacillus plantarum* + SAMe: 200 mg/day Comparator: placebo	Mild-to-moderate depression	Zung Self-Rating Depression Scale
Sakurai (2020) [36] RCT|United States n = 189|6 weeks	SAMe: 1600 mg/day for 6 weeks (non-responders: 3200 mg/day for 6 weeks) Escitalopram: 10 mg/day Comparator: placebo	Major Depressive Disorder	HAM-D, IDS-SR, CGI-S, CGI-I
Sarris (2014) [40] RCT|Australia n = 144|12 weeks	SAMe: 1600–3200 mg/day Escitalopram: 10 mg/day Comparator: placebo	Major Depressive Disorder	HAM-D
Sarris (2015) [41] RCT Re-Analysis|United States n = 189|12 weeks	SAMe: 1600–3200 mg/day Escitalopram: 10–20 mg/day Comparator: placebo	Major Depressive Disorder	HAM-D
Sarris (2018) [37] RCT|Australia n = 107 (77 completed)|8 weeks	SAMe: 800 mg/day Comparator: placebo	Non-remittent Major Depressive Disorder	MADRS
Sarris (2019) [38] RCT|Australia n = 158 | 8 weeks	SAMe 800 mg + folinic acid + Omega-3 fatty acids + 5-HTP + Zinc picolinate + relevant co-factors/day Comparator: placebo	Major Depressive Disorder	MADRS
Sarris (2020) [39] RCT|Australia n = 49 (41 completed)|8 weeks	SAMe: 800 mg/day Comparator: placebo	Major Depressive Disorder with mild-to-moderate symptoms	MADRS
Shippy (2004) [42] Non-randomized experimental study|United States n = 20 (15 completed)|8 weeks	1000 μg Vit B12 + 800 mg Folic Acid + SAMe: 400 mg/day (200 mg bid) increased to 1600 mg/day (800 mg bid) Comparator: None	Major Depressive Disorder	HAM-D (Response: ≥50% reduction in scores; Remission: HAM-D ≤ 7)
Strous (2009) [43] RCT|Israel n = 18 (15 completed)|8 weeks	SAMe: Week 1: 400 mg/day Weeks 2–8: 800 mg/day Comparator: placebo	Schizophrenia	PANSS, SANS, CGI, OAS, LHA, QLS
Targum (2018) [44] RCT|United States n = 234|8 weeks	SAMe:800 mg/day Comparator: placebo	Major Depressive Disorder	HAM-D, MADRS, IDS-SR
Targum (2020) [45] RCT (re-analysis)|United States n = 336|8 weeks	SAMe:800 mg/day Comparator: placebo	Major Depressive Disorder	HAM-D, MADRS, IDS-SR, CGI-S
Ullah (2022) [46] RCT (Crossover)|Europe n = 80 (65 completed)|3 months each	Crossover between: 200 mg/day SAMe + lactobacillus and placebo	Subthreshold depression Mild-to-moderate depression	HAM-D, PHQ-9

Scales: Hamilton Rating Scale for Depression (HAM-D), Beck Depression Inventory (BDI) and BDI Version II (BDI-II), Mini International Neuropsychiatric Interview (ICD-DSM MINI), Depression Anxiety Stress Scale (DASS), Structured Clinical Interview for DSM (SCID), Outcome Questionnaire 45 (OQ45), Warwick–Edinburgh Mental Well-being Scale (WBS), Quality of Life Scale (QOLS), Clinical Global Impression of Improvement (CGI-I), Clinical Global Impression–Severity scale (CGI-S), Kellner Symptom Questionnaire (SQ), Snaith–Hamilton Pleasure Scale (SHAPS), Sheehan Disability Scale (SDS), Positive and Negative Syndrome Scale (PANSS), Young Mania Rating Scale (YMRS), Children’s Depression Rating Scale–Revised (CDRS-R); ADHD Rating Scale IV (ADHD-RS), Beck Anxiety Inventory (BAI), Schwartz Outcome Scale (SOS-10), cognitive and physical symptoms questionnaire (CPFQ), Montgomery–Asberg Depression Rating Scale (MADRS), Inventory of Depressive Symptomatology–Self Rated (IDS-SR), Calgary Scale for Depression in Schizophrenia (SANS), Overt Aggression Scale (OAS), Life History of Aggression Scale (LHA), Quality of Life Scale (QLS), Patient Health Questionnaire-9 (PHQ-9) Other Acronyms: Systematic Review (SR), Meta-Analysis (MA), Randomized Controlled Trial (RCT), Attention Deficit/Hyperactivity Disorder (ADHD).

**Table 3 nutrients-16-03148-t003:** Study efficacy outcomes for studies examining the use of SAMe in CNS signs.

First Author (Year)	Efficacy
Alpert (2004) [14]	SAMe: Intent-to-treat analyses based on the HAM-D Response rate of 50% Remission rate of 43% Other improvements with SAMe: Significant decrease in Clinical Global Impressions Severity scores (4.0 ± 0.7 to 2.7 ± 1.2, *df* = 29; *p* < 0.0001) Significant improvement SQ-depression (13.2 ± 4.9 vs. 7.5 ± 5.3, respectively, *t*-test; *df* = 29; *p* = 0.001) and anxiety scores (12.2 ± 4.9, vs. 8.0 ± 5.6, respectively; *df* = 29; *p* = 0.012)
Anderson (2016) [15]	Anxiety levels improved Patient’s dose was titrated and continual improvement was shown
Arnold (2005) [16]	1600 mg more effective on CNS than 400 mg Both superior to placebo
Bambling (2015) [17]	1600 mg and 800 mg SAMe were effective: 35% achieved significant symptom improvement at the end of 15 weeks No significant difference between doses (*p* > 0.05) Non-responders saw improvements with adding 1600 mg of magnesium orotate BDI change: significant reduction in symptom scores [*df* = 18; *p* < 0.001] OQ45 change: significant reduction in functional distress scores [*df* = 18; *p* < 0.001] QOL change: significant increase in scores [*df* = 18; *p* < 0.001]
Carpenter (2011) [18]	Positive Results in Mild-to-Moderate (n = 9 studies): 5 studies reported significant positive results with SAMe in mild-moderate depression 2 studies reported positive results but not significant Mean treatment effect size = 1.0 (range 0.33–1.6) Positive Results in Moderate-to-Severe (n = 5 studies): 4 studies reported positive results but not significant Mean treatment effect size = 0.87 (median 0.79)
Cuomo 2020 [7]	SAMe (monotherapy) versus placebo: 3 out of 5 studies showed improvement SAMe (monotherapy) versus antidepressants (imipramine, escitalopram): 4 studies, no differences SAMe + SSRI vs. SSRI + placebo: 1 study, improvement
De Berardis (2013) [20]	Significant decrease in HAM-D score Response achieved by 60% (50% reduction in HAM-D score and CGI-I score of 1 or 2) Remission achieved by 36% (HAM-D score of ≤7) Significant reduction in SHAPS and SDS
Di Pierro (2015) [21]	No improvement for either group at 3 months Effectiveness demonstrated at 6 and 12 months for both groups SAMe vs. amitriptyline: SAMe had better results in terms of score, number of individuals in remission Amitriptyline at 6 and 12 months: score reduction was about 22% and 17% for the amitriptyline group SAMe at 6 and 12 months: score reduction was about 34% and 37%
Djokic (2017) [22]	Significant differences between SAMe and placebo in HAM-D and CGI-S scores at 3 months (*p* < 0.001) In SAMe: HAM-D improved from 20.17 ± 3.89 at baseline to 10.73 ± 3.4 with no influence of age or gender CGI-S improved from 4.1 ± 0.71 to 2.67 ± 0.76 CGI-I improved to 2.50 ± 0.68 (baseline not given)
Dolcetta (2013) [23]	4 patients tolerated the full dose and demonstrated efficacy: Behavioral improvements at 1 month and 12 months (n = 1) Improvement in sleep quality (n = 1) Improvement in self-injurious behavior and able to attend high school (n = 1) Improvement in anxiety and speech (n = 1)
Galizia (2016) [24]	No significant difference in depressive symptoms between SAMe and: escitalopram (n = 1), placebo (n = 2) Similar improvements in depressive symptoms, but no difference between SAMe and imipramine (n = 4) Significant improvement of SAMe vs. placebo when added on to an SSRI (*p* = 0.01; 73 participants; 1 study)
Green (2012) [25]	No treatment effect was found on ADHD symptoms (ADHD-RS) (*p* = 0.6) CGI-I improved in (28.6% SAMe vs. 14.3% placebo) Depressive disorder: SAMe group had a larger improvement on clinical scales than the placebo group
Jaggumantri (2015) [26]	Patient 1: SAMe was started at 50 mg/kg/day simultaneously with creatine, glycine, and L-arginine supplements Initial improved speech and ability to interact Stopped treatment at 3 months Restarted at 200 mg three times daily, which was reduced to 200 mg twice daily due to symptoms Patient 2: On SAMe 50 mg/kg/day, quality of speech and communication improved Discontinued after a few weeks
Kalman (2015) [27]	SAMe significantly: Reduced BDI-II scores at weeks four and eight, respectively (*p* = 0.001 and *p* = 0.002). Affected the BAI at week eight (*p* = 0.026). Changed the SOS-10 score at week eight (*p* = 0.038)
Levkovitz (2012) [28]	SAMe: Significantly improved ability to recall information (*p* = 0.04) Trend toward improved word-finding ability; *p* = 0.09
Limveeraprajak (2024) [5]	SAMe superior to placebo (SMD = −0.58, 95% CI = −0.93 to −0.23, *I*^2^ = 68%), even when two trials with a high risk of bias were excluded (SMD = −0.61, 91% CI = −1.05 to −0.17, *I*^2^ = 74%) IV route worse than IM or oral Moderate-certainty evidence suggests that SAMe may offer a moderate treatment effect SAMe + antidepressant vs. placebo + antidepressant: No significant difference between groups Very low-certainty evidence suggests that SAMe may not provide greater reduction in symptoms SAMe vs. antidepressant: No significant difference between groups (SMD = −0.22, 95% CI = −0.63 to 0.19, *I*^2^ = 76%) Low-certainty evidence suggests that SAMe may be as effective as antidepressants
Mischoulon (2012) [29]	SAMe: Plasma SAMe (*p* = 0.002) and SAH (*p* < 0.0001) levels significantly increased No change in HAM-D scores
Mischoulon (2014) [30]	SAMe vs. escitalopram vs. placebo: All treatment arms had significant reduction in HAM-D scores (*p* < 0.001) No significant difference between groups (*p* > 0.05) Response (50% decrease in HAM-D): not significant SAMe: 35% Escitalopram: 34% Placebo: 30% Remission (HAM-D score of ≤7): not significant SAMe: 28% Escitalopram: 28% Placebo: 17%
Murphy (2014) [31]	SAMe vs. placebo: No statistically significant differences in MADRS (*p* = 1.0), HAM-D (*p* = 1.0), or YMRS (*p* = 0.32) Model-estimated mean ratings at end point were 0.2 points higher for MADRS (corrected 95% CI = −11.1 to 11.5), 2.5 points lower for HAM-D (corrected 95% CI = −8.7 to 13.7), and 2.1 points higher for YMRS (95% CI = −2.3 to 6.5) for SAMe than for placebo
Papakostas (2010) [33]	SAMe + antidepressant vs. placebo + antidepressant: Nearly significant difference (*p* = 0.1) for lower endpoint HAM–D scores among SAMe-treated patients (mean: 11.1 [SD = 6.1]) relative to placebo-treated patients (mean: 15.8 [SD = 6.2]) Both outcome measures statistically significant in favor of adjunctive SAMe versus placebo (*p* = 0.01 and *p* = 0.02, respectively) Remission (HAM-D ≤7 or CGI-S = 1): significant (*p* = 0.03) SAMe = 10/39 Placebo = 2/34 Response (HAM-D 50% reduction or CGI-S < 3): significant (*p* = 0.02) SAMe = 20/39 Placebo = 7/34
Peng (2024) [34]	SAMe vs. placebo: No significant difference as monotherapy (SMD = –0.31, 95% CI = −0.65 to 0.03, *p* = 0.08; *I*^2^ = 66%) No significant difference in superiority as monotherapy (RR = 1.0, 95% CI = 0.79 to 1.26, *p* = 0.98; *I*^2^ = 0%) SAMe vs. imipramine or escitalopram No significant difference as monotherapy (SMD = 0.04, 95% CI = −0.09 to 0.16, *p* =0.56; *I*^2^ = 4%) No significant difference in superiority as monotherapy (RR = 1.08, 95% CI = 0.95 to 1.24, *p* = 0.24; *I*^2^ = 9%) SAMe vs. placebo as adjunctive therapy No significant difference (SMD = 0.02, 95% CI = −1.44 to 1.48, *p* = 0.98; *I*^2^ = 92%)
Saccarello (2020) [35]	SAMe + *Lactobacillus plantarum* vs. placebo: Significant improvement in depressive symptoms for treatment group (*p* = 0.0247) Significant improvement in absolute reduction in Zung score on day 14 for treatment (*p* = 0.0345) Significant improvement in cognitive and anxiety for treatment on day 14 (*p* = 0.0133)
Sakurai (2020) [36]	SAMe vs. escitalopram vs. placebo: No within-group and between-group differences were found in any of the efficacy measures for responders and non-responders
Sarris (2014) [40]	SAMe vs. escitalopram vs. placebo: All treatments had significant reduction in HAM-D score (F_1,100_ = 5.50, *p* = 0.021) Mean (SD) reduction: SAMe = 7.31 (5.96), escitalopram = 6.69 (5.70), placebo = 4.00 (5.64) Significant difference in improvement between groups (*p* = 0.039) SAMe vs. placebo: SAMe more effective Effect size for SAMe vs. placebo was moderate to large (d = 0.74) Significant effect between baseline and endpoint (F_1,65_ = 5.89, *p* = 0.018), with this effect occurring at every time point from week 1 (*p* = 0.04) to week 12 (*p* = 0.007) Higher remission rates (*p* = 0.003) SAMe vs. escitalopram: SAMe superior during weeks 2, 4, and 6 Remission rates (HAM-D < 7): Significantly different between groups (χ2_2,102_ = 8.57; *p* = 0.014) SAMe = 34% for SAMe Escitalopram = 23% Placebo = 6%
Sarris (2015) [41]	SAMe vs. escitalopram vs. placebo in HAM-D: Significant reduction for males treated with SAMe (−8.9 points) vs. placebo (−4.6 points) (*p* = 0.034) No other differences
Sarris (2018) [37]	SAMe + antidepressant vs. placebo + antidepressant: All groups had a significant reduction in MADRS over time (*p* < 0.001) Response rates (MADRS decrease of −50%) at week 8 (54.3% SAMe, 50% placebo, *p* = 0.68 between groups) Remission rates (MADRS score < 10 at final assessment) (43.5% SAMe, 38.3% placebo, *p* = 0.61 between)
Sarris (2019) [38]	Nutraceutical product vs. placebo: Placebo had a greater reduction in MADRS score (−1.75 points lower) Both groups improved over time No other significant differences between groups
Sarris (2020) [39]	SAMe: Clinically significant reduction in MADRS (−3.76 points) at 8 weeks Not statistically significant reduction in MADRS score between groups (*p* = 0.13)
Shippy (2004) [42]	Significant reduction in HAM-D scores, *df* = 13, *p* < 0.001 Response rate (intent-to-treat, all 19 patients): 74% Remission rate: 93% (14/15 patients who completed)
Strous (2009) [43]	SAMe vs. placebo: Significant improvements in SAMe patients only: Decrease in OAS scores (F = 5.6, *df* = 14, *p* = 0.032) Reduction of CGI-S scores (*p* < 0.01) Improvement in QLS scores (*p* < 0.001) Both groups, but greater reduction in SAMe: Improvement in CGI-I scores (*p* < 0.01)
Targum (2018) [44]	SAMe + antidepressant or placebo + antidepressant: No statistically significant treatment differences Note: study did not achieve primary endpoint due to subject selection differences First half of the study participants: favored SAMe Non-significant trend for the MADRS improvement (1st half: 27.8 ± 5.70, 2nd: 28.9 ± 6.65; *p* = 0.19) Non-significant trend for the IDS-SR improvement (*p* = 0.08)
Targum (2020) [45]	MADRS and HAM-D: SAMe was significantly better than placebo (F = 6.39; *df* = 1; *p* = 0.012), effect size = 0.404
Ullah (2022) [46]	Subjects who received the treatment in the placebo–SAMe order had higher PHQ-9 score values compared to those who received the treatment in a SAMe–placebo sequence (*p* = 0.030) HAM-D score decreased significantly between t0 and t1 measurements (*p* < 0.001) for all patients

Acronyms: Standardized Mean Difference (SMD).

**Table 4 nutrients-16-03148-t004:** Study safety outcomes for studies examining the use of SAMe in CNS signs.

First Author (Year)	Safety
Abeysundera (2018) [13]	Conclusion was that the patient experienced substance-/medication-induced mood disorder (when adding SAMe to the SSRI)
Alpert (2004) [14]	GI and headache side effects were most common No significant changes in weight, folate, B12, or homocysteine levels
Arnold (2005) [16]	Good tolerability
Bambling (2015) [17]	10 subjects dropped out
Carpenter (2011) [18]	Few instances of behavioral-related adverse events (AEs) No evidence for the occurrence of treatment-emergent suicidality detected AEs of mania/hypomania or psychomotor excitation following SAMe administration (n = 4) Increased anxiety-related AEs for SAMe than for placebo (n = 2)
Chitiva (2012) [19]	Patient attempted suicide after taking SAMe for 4 days
Cuomo 2020 [7]	Mild, transient, non-relevant side effects
De Berardis (2013) [20]	SAMe was well tolerated Most common adverse events: constipation (24%) nausea with decreased appetite (12%)
Di Pierro (2015) [21]	SAMe group had fewer side effects
Dolcetta (2013) [23]	Excess of excitement experienced at lower dosage, which led to discontinuations Increase in anxiety (n = 7)
Green (2012) [25]	No manic or psychotic symptoms No significant differences in side effects between groups Most common side effects were GI symptoms
Jaggumantri (2015) [26]	Patient 1: Discontinued 50 mg/kg/day at 3 months because of sleep disruptions and behavior issues 17 mg/kg/day was well-tolerated Patient 2: Discontinued 50 mg/kg/day because of behavioral issues
Kalman (2015) [27]	No significant adverse events
Limveeraprajak (2024) [5]	Generally well-tolerated
Mischoulon (2014) [30]	No significant differences in side effects between groups (*p* > 0.05) SAMe: GI, stomach discomfort, diarrhea
Murphy (2014) [31]	Discontinued after the 800 mg/day SAMe dosage due to brief episode of auditory hallucinations (n = 1) No other issues, including mania
Olsufka (2017) [32]	Treatment-emergent hypomania due to use of SAMe Admitted to the psych ER after “nervous breakdown” signs, racing thoughts, pressured speech Spouse described him as hyperactive, impulsive, loquacious, and irrational with side-to-side ocular movements
Papakostas (2010) [33]	No serious adverse events
Peng (2024) [34]	No significant difference between dropouts due to adverse effects (RR: 0.92, 95% CI: 0.49 to 1.73)
Saccarello (2020) [35]	Limited adverse events, which researchers believed were not related to products
Sakurai (2020) [36]	3200 mg/day SAMe: 31.3% experienced stomach or abdominal discomfort, significantly higher (*p* = 0.026) 25% experienced fluid retention or swelling (25.0%)
Sarris (2014) [40]	Well-tolerated No significant adverse events
Sarris (2018) [37]	5 SAMe group withdrawals possibly related to treatment: nausea, heightened anxiety, sleep issues
Sarris (2019) [38]	No significant differences between groups in adverse events More early termination in nutraceutical (n = 9) vs. placebo (n = 3); not significant
Sarris (2020) [39]	No significant differences in adverse events between groups (*p* = 0.53)
Shippy (2004) [42]	No dropouts due to side effects
Strous (2009) [43]	3 patients were discontinued from the study due to potential adverse effects of the study medication No significant differences between SAMe and placebo for all adverse events (all *p* > 0.05)
Targum (2018) [44]	High completion rate with 113 SAMe-assigned subjects (95.8%) Predominant adverse events were mild and primarily related to GI tract (<2% of patients)
Targum (2020) [45]	SAMe well-tolerated Predominant adverse events were mild and primarily related to GI tract

**Table 5 nutrients-16-03148-t005:** Quality assessment of CNS studies (n = 36 studies).

First Author Year	Clear Research Questions	Data Address Question	Total MMAT Score (out of 5)
Abeysundera 2018 [13]	Yes	Can’t Tell ^a^	4
Alpert 2004 [14]	Yes	Yes	3
Anderson 2016 [15]	No	No	3
Arnold 2005 [16]	Yes	Yes	5
Bambling 2015 [17]	Yes	Yes	4
Carpenter 2011 [18]	Yes	Yes	4
Chitiva 2012 [19]	Yes	Can’t Tell ^a^	4
Cuomo 2020 [7]	Yes	Yes	5
De Berardis 2013 [20]	Yes	Yes	5
Di Pierro 2015 [21]	Yes	Yes	4
Djokic 2017 [22]	Yes	Yes	5
Dolcetta 2013 [23]	Yes	Yes	5
Galizia 2016 [24]	Yes	Yes	5
Green 2012 [25]	Yes	Yes	4
Jaggumantri 2015 [26]	Yes	Can’t Tell ^a^	4
Kalman 2015 [27]	Yes	Yes	5
Levkovitz 2012 [28]	Yes	Yes	5
Limveeraprajak 2024 [5]	Yes	Yes	5
Mischoulon 2012 [29]	Yes	Yes	3
Mischoulon 2014 [30]	Yes	Yes	5
Murphy 2014 [31]	Yes	Yes	5
Olsufka 2017 [32]	Yes	Can’t Tell ^a^	4
Papakostas 2010 [33]	Yes	Yes	5
Peng 2024 [34]	Yes	Yes	5
Saccarello 2020 [35]	Yes	Yes	5
Sakurai 2020 [36]	Yes	Yes	5
Sarris 2014 [40]	Yes	Yes	5
Sarris 2015 [41]	Yes	Yes	5
Sarris 2018 [37]	Yes	Yes	5
Sarris 2019 [38]	Yes	Yes	5
Sarris 2020 [36]	Yes	Yes	5
Shippy 2004 [42]	Yes	Yes	5
Strous 2009 [43]	Yes	Yes	4
Targum 2018 [44]	Yes	No ^b^	3
Targum 2020 [45]	Yes	Yes	5
Ullah 2022 [46]	Yes	Yes	5

^a^ Case report, so there is limited evidence whether the data address the question. Further, the authors did not use a Naranjo scale to evaluate the likelihood that the medication caused the event, limiting the ability to evaluate this. However, the authors did use the literature and the case progression to substantiate the potential event. ^b^ Inclusion criteria resulted in differences between groups, limiting the validity of the results.

## Data Availability

No new data were created outside of what is published in this article. The study protocol, data collection forms, data extracted from included studies, and all other materials used in this review can be made available upon request. No amendments were made to the study protocol.

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
