# Peer review of "S-Adenosylmethionine (SAMe) for Central Nervous System Health: A Systematic Review"

_nutrients, 2024, doi:10.3390/nu16183148_

Round 1

Reviewer 1 Report

Comments and Suggestions for Authors

Comment-Nutrients

This review presents a systematic evaluation of the effects of S-Adenosylmethionine on the central nervous system, with some comments as follows:

1.      The conclusions obtained after the systematic review and the outlook or direction for future development should be added to the abstract.

2.      In the keywords section “S-adenosylmethionine; SAMe” don't these two mean the same thing?

3.      The introduction section does not clearly state why the author synthesizes the topic and the novelty of the review. After all, what are the strengths of this review in comparison to the total number of “S-Adenosylmethionine affects the nervous system”? These pieces of information need to be added in the introduction.

4.      All of the author's references are placed at the beginning of the sentence. Please standardize the formatting and citing of references.

5.      It is proposed to add a section describing current challenges and future perspectives.

6.      This article is too short on detailed explanations and descriptions of Tables 1 through 5, and it is recommended that additional explanations be added or subtracted.

7.      Most of the results in the article are presented in the form of tables, and it is suggested that the authors could have added a description of “S-Adenosylmethionine (SAMe) for Central Nervous System 2 Health” as appropriate employing a diagram.

Comments on the Quality of English Language

 Moderate editing of English language required.

Author Response

Comment 1: The conclusions obtained after the systematic review and the outlook or direction for future development should be added to the abstract. 

Response 1: We added a specific statement of future directions to the abstract and further specified the conclusions.

Comment 2: In the keywords section “S-adenosylmethionine; SAMe” don't these two mean the same thing?

Response 2: They do mean the same thing. We excluded the key word, “SAMe.”

Comment 3: The introduction section does not clearly state why the author synthesizes the topic and the novelty of the review. After all, what are the strengths of this review in comparison to the total number of “S-Adenosylmethionine affects the nervous system”? These pieces of information need to be added in the introduction. 

Response 3: We included the point in the introduction that an updated review of SAMe for specific CNS conditions is needed. In addition, this is also a novel review because we highlight potential dosing choices and strategies for these conditions. 

Comment 4: All of the author's references are placed at the beginning of the sentence. Please standardize the formatting and citing of references.

Response 4: Our references are numbered at the end of the sentence (after the period), in line with the Nutrients formatting. We apologize if that was unclear! 

Comment 5: It is proposed to add a section describing current challenges and future perspectives. 

Response 5: At the end of the discussion section, we added two paragraphs addressing both the current challenges (e.g. lack of long-term efficacy and safety past 12 months) and future directions (e.g. address differences in oral formulations to better recommend appropriate doses). 

Comment 6: This article is too short on detailed explanations and descriptions of Tables 1 through 5, and it is recommended that additional explanations be added or subtracted. Most of the results in the article are presented in the form of tables, and it is suggested that the authors could have added a description of “S-Adenosylmethionine (SAMe) for Central Nervous System 2 Health” as appropriate employing a diagram.

Response 6: Further explanations and key points for Tables 1-5 were added to the text of the results section.

Please see attachment to view another format of the comments and our responses.

Reviewer 2 Report

Comments and Suggestions for Authors

Baden and colleagues present a systematic review assessing the efficacy, safety, and optimal dosage of S-adenosylmethionine (SAMe) for mood-related and central nervous system (CNS)-related symptoms.

The use of a PRISMA-based systematic review approach ensures a rigorous and systematic process in identifying and reviewing relevant literature. The involvement of multiple researchers in the review and data extraction process, with a third researcher resolving conflicts, adds to the robustness of the study design.

The abstract mentions that SAMe was "often effective" but does not provide specific data or effect sizes from the studies reviewed. 

The mentions "limited or reasonable adverse events," do not specify what these events were, their frequency, or their severity. Given that safety is a critical aspect of evaluating any therapeutic intervention, a more detailed discussion of adverse events is necessary.

Furthermore, the optimal dose for specific CNS-related symptoms (e.g., depression vs. sleep disturbances) is not clearly discussed page 133, which is a missed opportunity to provide more tailored and discussed recommendations.

The conclusion that SAMe "may help manage depressive symptoms" is broad and does not adequately reflect the nuanced findings that might be present in the reviewed literature. More specific conclusions, based on the evidence reviewed, would be more informative.

This review doesn't provide a significant study or relevant news on the subject.

Author Response

Comment 1: The abstract mentions that SAMe was "often effective" but does not provide specific data or effect sizes from the studies reviewed.

Response 1: We included the data from our results “effective in 24 of 36 studies” to the abstract to provide a more specific description of the data. 

Comment 2: The mentions "limited or reasonable adverse events," do not specify what these events were, their frequency, or their severity. Given that safety is a critical aspect of evaluating any therapeutic intervention, a more detailed discussion of adverse events is necessary.

Response 2: In the abstract, we provided a more detailed description of our finding regarding the GI side effects. In addition, we added more explanation to the results and discussion sections on these side effects as well as all other reported side effects included in the studies. 

Comment 3: Furthermore, the optimal dose for specific CNS-related symptoms (e.g., depression vs. sleep disturbances) is not clearly discussed page 133, which is a missed opportunity to provide more
tailored and discussed recommendations.

Response 3: We have included a more robust discussion on optimal dose for specific CNS symptoms/diseases in the discussion. We highlight the most robust evidence is found for depression, while also recommending specific starting doses to consider.

Comment 4: The conclusion that SAMe "may help manage depressive symptoms" is broad and does not adequately reflect the nuanced findings that might be present in the reviewed literature. More specific conclusions, based on the evidence reviewed, would be more informative.

Response 4: In the conclusion, we specified that there were improvements in depressive symptoms with using SAMe as both a monotherapy and with concurrent antidepressant or nutraceutical therapies.

Comment 5: This review doesn't provide a significant study or relevant news on the subject.

Response 5: We believe this review is a significant study because it provides updated evidence/current knowledge on SAMe in CNS conditions. We highlight several key points, including how its substantial benefit is for depressive symptoms (over other CNS conditions), most common as well as all reported side effects from included studies, as well as several recommendations on optimal doses based on the CNS condition. This is all important information for clinicians as they seek to have accurate information and provide optimal recommendations on SAMe for their patients.

Please see attachment for another format of reviewing the comments and our responses.

Round 2

Reviewer 1 Report

Comments and Suggestions for Authors

Author Response

See previous responses

Reviewer 2 Report

Comments and Suggestions for Authors

Although considering that in this article there are some issues that could be improved related to the research design, I believe that the modifications made turned the document better and can be published.

Author Response

See previous response.